# Protection against Lipopolysaccharide-Induced Endotoxemia by Terrein Is Mediated by Blocking Interleukin-1β and Interleukin-6 Production

**DOI:** 10.3390/ph15111429

**Published:** 2022-11-18

**Authors:** Yeo Dae Yoon, Myeong Youl Lee, Byeong Jo Choi, Chang Woo Lee, Hyunju Lee, Joo-Hee Kwon, Jeong-Wook Yang, Jong Soon Kang

**Affiliations:** Laboratory Animal Resource and Research Center, Korea Research Institute of Bioscience and Biotechnology, 30 Yeongudanjiro, Cheongju 28116, Republic of Korea

**Keywords:** terrein, endotoxemia, interleukin-1β, interleukin-6, NF-κB

## Abstract

Terrein is a fungal metabolite and has been known to exert anti-melanogenesis, anti-cancer, and anti-bacterial activities. However, its role in endotoxemia has never been investigated until now. In the present study, we examined the effect of terrein on lipopolysaccharide (LPS)-induced endotoxemia in mice and characterized the potential mechanisms of action. Treatment with terrein increased the survival of mice and decreased the production of inflammatory cytokines, including interleukin-1β (IL-1β) and interleukin-6 (IL-6) in an LPS-induced endotoxemia model. In addition, terrein suppressed the LPS-induced production of IL-1β and IL-6 in RAW 264.7 cells, a murine macrophage-like cell line, and the mRNA expression of IL-1β and IL-6 was also inhibited by terrein in LPS-stimulated RAW 264.7 cells. Further study demonstrated that terrein blocked LPS-induced phosphorylation of p65 subunit of nuclear factor (NF)/κB and the phosphorylation of c-Jun N-terminal kinase (JNK) and p38 mitogen-activated protein kinase (MAPK) was also suppressed by terrein treatment. Collectively, these results suggest that terrein exerts a protective effect again LPS-induced endotoxemia in mice by blocking the production of inflammatory cytokines. Our results also suggest that the anti-inflammatory effect of terrein might be mediated, at least in part, by blocking the activation of NF-κB, JNK, and p38 MAPK signaling pathways.

## 1. Introduction

Terrein is a bioactive metabolite with a phenolic structure isolated from a fungus *Aspergillus terreus* (Figure 1) [1,2]. The firstly identified biological activity of terrein was an anti-melanogenesis activity [3,4] and terrein has also been known to exert a variety of biological activities, including anti-cancer and anti-bacterial activities [5,6,7,8].

Sepsis is a dysregulated host response to infection and is caused by the excessive immune response to pathogens [9,10]. An exposure to lipopolysaccharide (LPS), the main structural component of a Gram-negative bacterial cell wall, is the most common cause of sepsis and induces uncontrolled inflammatory responses, hypotension, and vasoplesia leading to multiple organ dysfunction and ultimately death [11,12,13]. Bacterial LPS elicits the overproduction of inflammatory mediators, such as interleukin-1β (IL-1β), interleukin-6 (IL-6), tumor necrosis factor-α (TNF-α), nitric oxide, and prostaglandin E2 and the large number of inflammatory mediators produced in the body are thought to contribute to the LPS-induced symptoms of sepsis [14,15]. IL-1β is well-known to contribute to the development of sepsis and induces the production of other inflammatory mediators [15]. Mice treated with recombinant interleukin-1 receptor antagonist had improved survival in mouse models of endotoxemia [16]. It has also been reported that the treatment of patients with sepsis with human recombinant interleukin-1 receptor antagonist provided a dose-related survival advantage in clinical trials [17]. IL-6 is also implicated in the development of sepsis and IL-6 blockade has been shown to improve survival in a mouse model of sepsis [18,19]. In addition, it was reported that the high serum levels of IL-6 and IL-1 are associated with fatal outcome in meningococcal septic shock [20]. Therefore, these reports suggest that IL-1β and IL-6 are important inflammatory cytokines involved in the pathogenesis of sepsis.

The objective of this study was to examine the effect of terrein on LPS-induced endotoxemia in mice. We also briefly investigated the mechanisms responsible for the anti-septic and anti-inflammatory effects of terrein.

## 2. Results

### 2.1. Effect of Terrein on Lipopolysaccharide (LPS)-Induced Endotoxemia in C57BL/6 Mice

To investigate the effect of terrein on sepsis in vivo, we examined the effect of terrein on the LPS-induced endotoxemia. As shown in Figure 2, the administration of a high dose of LPS (50 mg/kg) resulted in 100% mortality within 48 h and the median survival time of the vehicle-treated group was 23 h. However, treatment with 10 mg/kg or 30 mg/kg of terrein increased the median survival time to 28 h (*p* = 0.2305) and 60 h (*p* = 0.0038), respectively (Figure 2).

To investigate whether the protective effect of terrein on LPS-induced endotoxemia is mediated by inhibiting inflammatory responses, we analyzed the serum levels of inflammatory cytokines, including IL-1β and IL-6. Figure 3A,B show that the serum levels of IL-1β and IL-6 were dramatically increased by LPS treatment. However, treatment with 30 mg/kg of terrein significantly suppressed the levels of inflammatory cytokines induced by LPS treatment (Figure 3A,B). In addition, we also examined the levels of IL-1β and IL-6 in lung tissue. As shown in Figure 3C,D, LPS treatment substantially increased the levels of IL-1β and IL-6 in the lungs and this was inhibited significantly by terrein (30 mg/kg) treatment.

### 2.2. Effect of Terrein on LPS-Induced Production of IL-1β and IL-6 in RAW 264.7 Cells

To further confirm the anti-inflammatory effect of terrein, we investigated the effect of terrein on the production of inflammatory cytokines in a mouse macrophage cell line, RAW 264.7. Figure 4A shows that terrein had no significant cytotoxic effect in LPS-stimulated RAW 264.7 cells at the concentrations used in this study. The treatment of RAW 264.7 cells with LPS (200 ng/mL) markedly induced the production of IL-1β and IL-6 (Figure 4B,C). However, terrein suppressed the LPS-induced production of IL-1β and IL-6 in a concentration-dependent manner and treatment with 30 μM of terrein resulted in 67% and 78% inhibition of IL-1β and IL-6 production, respectively, in LPS-stimulated RAW 264.7 cells (Figure 4B,C).

### 2.3. Effect of Terrein on LPS-Induced mRNA Expression of IL-1β and IL-6 in RAW 264.7 Cells

To further investigate whether the inhibitory effect of terrein on the production of IL-1β and IL-6 was due to the inhibitory effect of terrein on the mRNA expression of cognate genes, we examined the effect of terrein on LPS-induced mRNA expression of IL-1β and IL-6 in RAW 264.7 cells using quantitative RT-PCR. As shown in Figure 5A,B, LPS substantially increased the mRNA expression of IL-1β and IL-6. However, this induction was inhibited by terrein treatment in a dose-dependent manner (Figure 5A,B).

### 2.4. Effect of Terrein on LPS-Induced Activation of NF-κB and MAPK Signaling Pathways in RAW 264.7 Cells

To further investigate the molecular mechanisms responsible for the inhibitory effect of terrein on the gene expression of IL-1β and IL-6, we first evaluated the effect of terrein on LPS-induced phosphorylation of p65 subunit of NF-κB at S536 residue of transactivation domain, which leads to enhanced transactivation of NF-κB. As shown in Figure 6A, terrein dose-dependently inhibited the phosphorylation of the p65 subunit in LPS-stimulated RAW 264.7 cells. We also examined the effect of terrein on the phosphorylation of ERK, JNK, and p38 MAPK in LPS-stimulated RAW 264.7 cells. Figure 6B shows that treatment with terrein suppressed the LPS-induced phosphorylation of JNK and p38 MAPK, but not that of ERK, in RAW 264.7 cells.

## 3. Discussion

Although terrein was discovered in 1935, it has received little attention for a long time [21]. However, after the first report describing its biological activity as a melanogenesis inhibitor, terrein has drawn attention and studies that have shown it has a variety of biological activities, including anti-cancer, anti-microbial, and insecticidal activities [6,7,8,22,23,24]. In this study, we demonstrated that terrein has a protective effect against LPS-induced endotoxemia in mice and inhibits the production of inflammatory cytokines in vitro and in vivo, suggesting that terrein might be a therapeutic candidate for the treatment of sepsis or other inflammatory diseases.

Sepsis is a state of dysregulated inflammation and the development and progression of sepsis are complex and multi-factorial [25]. A variety of animal models were developed to create reproducible systems for studying sepsis and mouse models were often used due to the ease of experimentation, the availability of genetically engineered species, and the relatively low cost [26]. Among several mouse models of sepsis, the LPS-induced endotoxemia model is known as the most simple and reproducible way to re-capitulate human sepsis and has been used for nearly 100 years [26]. Therefore, we examined the anti-septic effect of terrein using the LPS-induced endotoxemia model. In this study, our results showed that terrein increased the median survival time of mice, suggesting that terrein has a protective effect against LPS-induced endotoxemia in mice.

IL-1β and IL-6 are well-known pro-inflammatory cytokines and the role of IL-1β and IL-6 in sepsis has been well documented previously [15,16,18,19]. Therefore, we investigated the effect of terrein on the serum levels of IL-1β and IL-6 in an LPS-induced endotoxemia model. Our results demonstrated that LPS-induced increase of IL-1β and IL-6 was significantly suppressed by terrein treatment. Acute lung injury is a life-threatening lung change and 40% of acute lung injury cases result from sepsis [27]. However, effective treatment for sepsis-induced acute lung injury and lung inflammation are limited [27]. In this study, our results showed that the levels of IL-1β and IL-6 in the lung were increased by LPS treatment and this was significantly inhibited by terrein treatment, suggesting that terrein might be a therapeutic candidate for the treatment of acute lung injury induced by sepsis. We also examined the effect of terrein on the production and mRNA expression of IL-1β and IL-6 in RAW 264.7 cells. Consistent with the results of in vivo experiments, the terrein concentration dependently suppressed the production of IL-1β and IL-6 in LPS-stimulated RAW 264.7 cells. In addition, the LPS-induced mRNA expression of IL-1β and IL-6 was also abrogated by terrein treatment, suggesting that the inhibitory effect of terrein on the production of IL-1β and IL-6 might be due, at least in part, to its effect on the mRNA expression of cognate genes. The various post-transcriptional and post-translational processes, including conversion of pro-IL-1β to active IL-1β, can be affected by terrein treatment and the effect of terrein on these processes might be investigated in the future.

Among various transcription factors involved in the expression of inflammatory genes, NF-κB is one of the most important transcription factors in terms of directing the transcription of inflammatory cytokines after exposure to bacterial LPS [25,28]. To elucidate the molecular mechanism responsible for the inhibitory effect of terrein on the mRNA expression of IL-1β and IL-6, we examined the effect of terrein on the LPS-induced phosphorylation of p65 subunit of NF-κB, a classic marker of NF-κB activation. Our results demonstrated that terrein inhibited the phosphorylation of the p65 subunit in a dose-dependent manner in LPS-stimulated RAW 264.7 cells. The MAPK signaling pathway has also been known to play important roles in the expression of IL-1β and IL-6 [29,30,31]. In accordance with our result, Lee and coworkers also showed the inhibitory effect of terrein on NF-κB activation in LPS-stimulated human dental pulp cells [23]. To further investigate, we also examined the effect of terrein on the activation of the MAPK signaling pathway. Treatment with LPS substantially increased the phosphorylation of MAPKs and terrein blocked the LPS-induced phosphorylation of JNK and p38 MAPK in RAW 264.7 cells. It was reported that the specific inhibitors of JNK and p38 MAPK suppressed LPS-induced expression of IL-1β and IL-6 in mouse macropahges, suggesting that the inhibition of JNK and p38 MAPK activation might be one of the mechanisms responsible for the inhibitory effect of terrein on the expression of IL-1β and IL-6 [32,33,34]. These results suggest that the inhibitory effect of terrein on the mRNA expression of IL-1β and IL-6 might be mediated by the downregulation of NF-κB, JNK, and p38 MAPK signaling pathways.

In summary, the results presented in this report demonstrated that terrein inhibits LPS-induced endotoxemia in mice and decreased levels of IL-1β and IL-6 in serum and lungs were observed in terrein-treated mice. Our results also showed that terrein suppresses the LPS-induced expression of IL-1β and IL-6 in mouse macrophage cell line and this is mediated, at least in part, by blocking the activation of NF-κB, JNK, and p38 MAPK signaling pathways. These results provided evidence for the possible application of terrein as a therapeutic candidate for the treatment of sepsis or other inflammatory diseases.

## 4. Materials and Methods

### 4.1. Mouse Model of Lipopolysaccharide-Induced Sepsis

Male C57BL/6 mice (7 weeks old) were purchased from Koatech (Pyungtaek, Gyeonggi, Republic of Korea) and allowed to acclimate for at least 1 week before use. The mice were randomly divided into three groups (*n* = 10 in each group) and treated intraperitoneally (i.p.) with vehicle (phosphate-buffered saline) or terrein (10 mg/kg or 30 mg/kg). After 2 h, the mice were treated with LPS (50 mg/kg, i.p., from *Escherichia Coli* O55:B5) and the survival of the mice was monitored for 120 h. For cytokine analysis, blood and lung tissues were collected 2 h after LPS treatment.

### 4.2. Chemicals and Cell Culture

All reagents were purchased from Sigma-Aldrich (St Louis, MO, USA) unless otherwise stated. RAW 264.7 cells (ATCC TIB71) were grown in Dulbecco’s modified Eagle’s medium (Invitrogen, Carlsbad, CA, USA), supplemented with 10% fetal bovine serum, 2 mM L-glutamine, 100 U/mL penicillin, and 100 mg/mL streptomycin at 37 °C in 5% CO_2_ humidified air.

### 4.3. Cell Viability Assay

The cell viability assay was performed using a Cell Proliferation Kit II (Roche Applied Science, Mannheim, Germany) according to the manufacturer’s instructions. Briefly, the XTT labeling mixture was prepared by mixing 50 volumes of 1 mg/mL sodium 3′-[1-(phenylaminocarbonyl)-3,4-tetrazolium]-bis(4-methoxy-6-nitro)benzene sulfonic acid hydrate with 1 volume of 0.383 mg/mL of N-methyldibenzopyrazine methyl sulfate, added to the cultures and incubated for 2 h at 37 °C. The absorbance was measured at 495 nm with a reference wavelength of 650 nm.

### 4.4. Enzyme-Linked Immunosorbent Assay (ELISA)

The RAW 264.7 cells were plated at 5 × 10^5^ cells/mL and stimulated with LPS (200 ng/mL) in the presence or absence of terrein (1, 3, 10, or 30 μM) for 24 h. The culture supernatants were collected and the amount of cytokine was determined using a mouse IL-1β Quantikine ELISA kit or mouse IL-6 Quantikine ELISA kit (R&D Systems, Inc., Minneapolis, MN, USA) according to the manufacturer’s instructions.

### 4.5. RNA Isolation and Quantification of mRNA Expression

The total cellular RNA was extracted using RNeasy Plus Mini Kit (Qiagen, Valencia, CA, USA) with RNase-Free DNase Set (Qiagen) according to the manufacturer’s instructions. Equal amounts of RNA were reverse transcribed into cDNA by using oligo(dT)_15_ primers and the resulting cDNA was amplified by qPCR using Power SYBR Green PCR Master Mix (Invitrogen). The samples were amplified by 45 cycles of denaturation (95 °C for 15 s) and amplification (60 °C for 1 min using ABI 7500 Sequence Detection System (Applied Biosciences, Foster City, CA, USA). The relative gene expression levels relative to the control gene (β-actin) were calculated using the 2^−∆∆Ct^ method. The primer sequences used were as follows: mouse IL-1β, sense 5′-TGC AGA GTT CCT ACA TGG TCA ACC C-3′, antisense 5′-GTG CTG CCT AAT GTC CCC TTG AAT C-3′; mouse IL-6, sense 5′-TGC TGG TGA CAA CCA CGG CC-3′, antisense 5′-GTA CTC CAG AAG ACC AGA GG-3′; and mouse β-actin, sense 5′-TGG AAT CCT GTG GCA TCC ATG AAA C-3′, antisense 5′-TAA AAC GCA GCT CAG TAA CAG TCC G-3′.

### 4.6. Western Immunoblot Analysis

The total protein extracts were prepared by lysing cells in Cell Lysis Buffer (Cell Signaling Technology, Beverly, MA, USA) with a protease inhibitor cocktail (Merck Millipore, Billerica, MA, USA) and phosphatase inhibitors (Sigma-Aldrich). The protein concentrations in the lysates were determined using a BCA Protein Assay Kit (Pierce Biotechnology, Waltham, MA, USA) according to the manufacturer’s instructions. The protein extracts were separated using sodium dodecyl sulfate-polyacrylamide gel electrophoresis and transferred to nitrocellulose membranes. The membranes were incubated with a blocking buffer (tris-buffered saline containing 0.05% tween 20 and 5% non-fat dried milk) and probed with the primary antibodies against p-ERK1/2, ERK1/2, p-SAPK/JNK, SAPK/JNK, p-p38, p38, p-p65, p65, p-IκBα, or IκBα (Cell Signaling Technology). After washing, the membranes were probed with horseradish peroxidase (HRP)-conjugated secondary antibodies and visualized using an Immobilon Western Chemiluminescent HRP substrate (Merck Millipore).

### 4.7. Statistical Analysis

The mean ± S.D. was determined for each treatment group in each experiment. The data were analyzed by the analysis of variance and Dunnett’s multiple comparison test by using Prism (GraphPad Software Inc., San Diego, CA, USA) software. For the survival study, the Log-rank test was used for statistical analysis. The criterion for statistical significance was set at *p* < 0.05.

## Figures and Tables

**Figure 1 pharmaceuticals-15-01429-f001:**
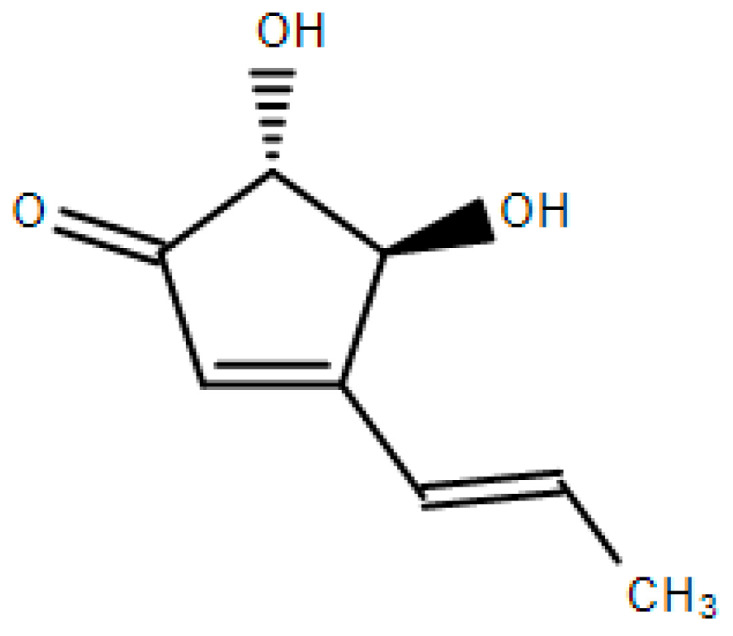
Chemical structure of terrein.

**Figure 2 pharmaceuticals-15-01429-f002:**
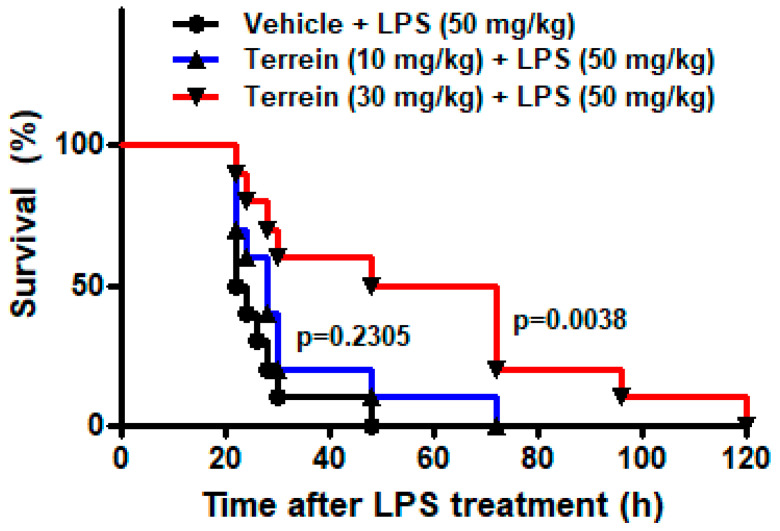
Effect of terrein on survival of mice in LPS−induced endotoxemia model. Different groups (*n* = 10) of animals were treated with vehicle or indicated concentrations of terrein 2 h before LPS (50 mg/kg) treatment. The survival rate was monitored until 120 h.

**Figure 3 pharmaceuticals-15-01429-f003:**
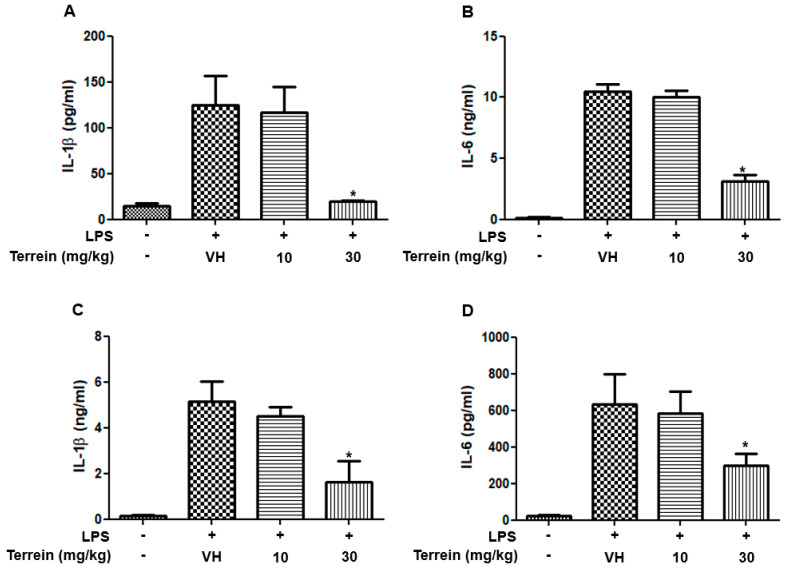
Effect of terrein on the production of inflammatory cytokines in LPS−treated mice. Different groups (*n* = 10) of animals were treated with vehicle or indicated concentrations of terrein 2 h before LPS (50 mg/kg) treatment. After 2h, mice were sacrificed and the serum levels of IL−1β (**A**), IL−6 (**B**), the lung tissue levels of IL−1β (**C**), and IL−6 (**D**) were measured using ELISA. Each column shows the mean ± standard deviation of 10 mice. Significance was determined using Dunnett’s *t*−test versus the vehicle−treated group (* *p* < 0.05).

**Figure 4 pharmaceuticals-15-01429-f004:**
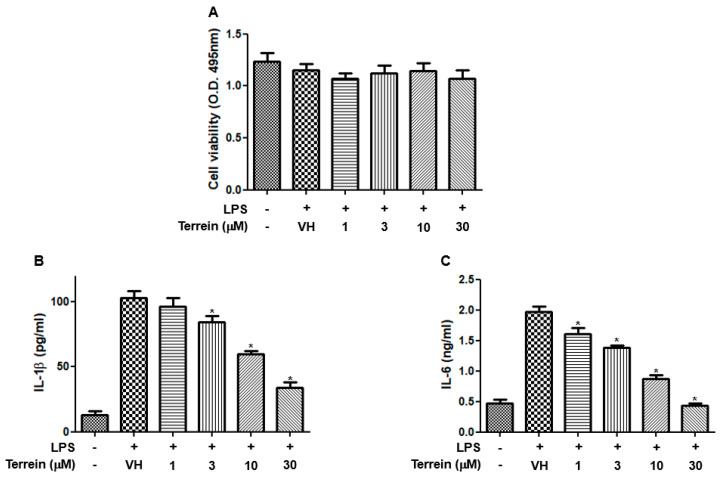
Effect of terrein on LPS−induced production of inflammatory cytokines in RAW 264.7 cells. RAW 264.7 cells were pretreated with the indicated concentrations of terrein for 1 h before being incubated with LPS (200 ng/mL) for 24 h. (**A**) Cell viability was analyzed using XTT assay. The production of IL−1β (**B**) and IL−6 (**C**) were measured using ELISA. Each column shows the mean ± standard deviation of quadruplicate determinations. Significance was determined using Dunnett’s *t*−test versus vehicle−treated group (* *p* < 0.05).

**Figure 5 pharmaceuticals-15-01429-f005:**
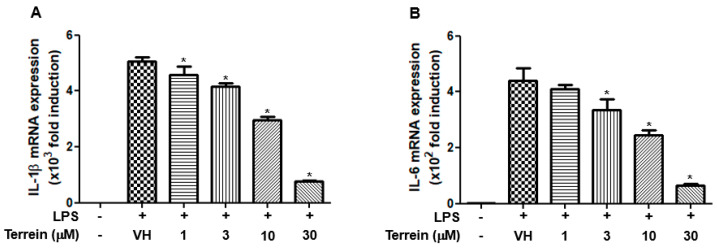
Effect of terrein on LPS−induced mRNA expression of inflammatory cytokines in RAW 264.7 cells. RAW 264.7 cells were pretreated with the indicated concentrations of terrein for 1 h before being incubated with LPS (200 ng/mL) for 6 h. The mRNA expression of IL−1β (**A**) and IL−6 (**B**) was measured using quantitative RT−PCR. Each column shows the mean ± standard deviation of quadruplicate determinations. Significance was determined using Dunnett’s *t*−test versus vehicle−treated group (* *p* < 0.05).

**Figure 6 pharmaceuticals-15-01429-f006:**
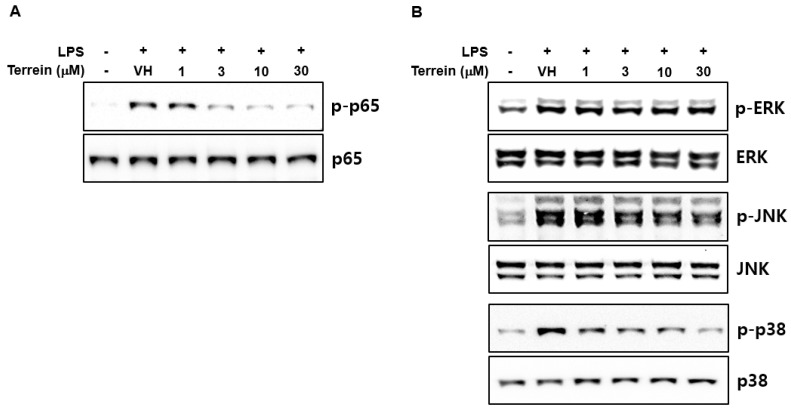
Effect of terrein on the activation of NF−κB and MAPK signaling pathway. RAW 264.7 cells were pretreated with the indicated concentrations of terrein for 1 h before being incubated with LPS (200 ng/mL) for 1 h (for p65) or 10 min (for MAPKs). The phosphorylation of p65 (**A**) and MAPKs (**B**) was examined using Western immunoblot analysis.

## Data Availability

The data is contained in the article.

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
