# Peer review of "Protection against Lipopolysaccharide-Induced Endotoxemia by Terrein Is Mediated by Blocking Interleukin-1β and Interleukin-6 Production"

_pharmaceuticals, 2022, doi:10.3390/ph15111429_

Round 1

Reviewer 1 Report

Authors explored the potential effect of terrein on lipopolysaccharide-induced endotoxemia in a mouse model and inflammatory mediators in RAW 264.7 cells. Although authors showed some significant effects of terrein, there are still some critical issues which were not addressed.  

1.      LPS-induced endoxemia and subsequent death of mice can involve various cytokines/chemokines. Authors did not provide sufficient evidence or rationales to convince us the critical or indispensable roles of IL-1β and IL-6 in this model.

How about the roles of cytokines/chemokines other than IL-1β/IL-6 in this model?

Can blockade of IL-1β/IL-6 affect the survival in the mouse model?

2.      I wonder if the lung damage was the leading cause of death in the mouse model.   

3.      Could the suppression of p-38 and Jnk phosphorylation account for the reduction of LPS-induced IL-1β/IL-6 induction?

4.      Was the activation of IL-1β proforms affected by terrein? Or the expression of IL-1β proform was affected?

Reviewer 2 Report

This is the first study to address the in vivo protective anti-inflammatory properties of terrein. It provides evidence of strong anti-endotoxemia and pro-survival effects of terrein in mice, that show promise in clinical settings. Overall, the manuscript is concise and results clearly presented. It should be stated in discussion that NFκΒ inhibition by terrein has been already been demonstrated in Lee et. al, 2008, in support to the present observation that terrein inhibits the LPS-induced phosphorylation of p65 subunit of NFκΒ.

Author Response

This is the first study to address the in vivo protective anti-inflammatory properties of terrein. It provides evidence of strong anti-endotoxemia and pro-survival effects of terrein in mice, that show promise in clinical settings. Overall, the manuscript is concise and results clearly presented. It should be stated in discussion that NFkB inhibition by terrein has been already been demonstrated in Lee et. al. 2008, in support to the present observation that terrein inhibits the LPS-induced phosphorylation of p65 subunit of NFkB

- According to reviewers suggestion, we stated the inhibitory effect of terrein on NF-kB in LPS-stimulated human dental pulp cells reported by Lee et. al. 2008 in discussion section of revised manuscript.

Round 2

Reviewer 1 Report

In figure 5, inhibition of IL-1beta mRNA expression by terrein did not mean formation of active IL-1beta was suppressed by terrein. That should be clarified.
